# Decrease in the prevalence of antimicrobial resistance in *Escherichia coli* isolates of Canadian turkey flocks driven by the implementation of an antimicrobial stewardship program

Rima D. Shrestha[1]*, Agnes Agunos[2], Sheryl P. Gow[3], Anne E. Deckert[2], Csaba Varga[1,4]*

1 Department of Pathobiology, College of Veterinary Medicine, University of Illinois at Urbana-Champaign, Urbana, Illinois, United States of America, 2 Center for Foodborne, Environmental and Zoonotic Infectious Diseases, Public Health Agency of Canada, Guelph, Ontario, Canada, 3 Center for Foodborne, Environmental and Zoonotic Infectious Diseases, Public Health Agency of Canada, Saskatoon, Saskatchewan, Canada, 4 Carl R. Woese Institute for Genomic Biology, University of Illinois Urbana-Champaign, Urbana, Illinois, United States of America

* cvarga@illinois.edu (CV); rdshrest@illinois.edu (RDS)

**Data Availability Statement:** All relevant data are within the paper and its Supporting information files.

## Abstract

The emergence of antimicrobial-resistant organisms at the human-animal-environment interface has raised global concern prompting governments and various stakeholders to take action. As a part of the stewardship initiative, Canadian turkey producers have implemented an antimicrobial use (AMU) strategy to manage antimicrobial resistance (AMR) in their sector. This study evaluated farm-level AMU and AMR data collected between 2016 and 2021 in major turkey-producing provinces/regions through the Canadian Integrated Program for Antimicrobial Resistance Surveillance to assess the progress of the strategy by characterizing the prevalence of homologous and multidrug resistance (MDR) in *Escherichia coli* isolated from turkeys. Multivariable mixed-effect logistic regression models assessed temporal and provincial/regional variations in AMR and MDR. Negative binomial regression models examined the temporal and regional variations in the total AMU. The total AMU (measured in mg/kg turkey biomass) significantly decreased in all provinces/regions in 2020 and 2021. *Escherichia coli* isolates from turkey flocks showed a significant decrease in resistance to gentamicin, sulfisoxazole, and tetracyclines during the six-year study period, consistent with the timing of the AMU reduction strategy. The prevalence of MDR isolates was significantly lower in 2020 and 2021 compared to 2016. Higher prevalence was observed in the Western region compared to Québec and Ontario. Two common AMR patterns were identified: ampicillin-streptomycin-tetracyclines and streptomycin-sulfisoxazole-tetracyclines. These AMR patterns indicate possible cross-resistances (same class), co-selection (unrelated classes) for resistance, or potential carryover of resistance determinants from previous production cycles. The decreasing prevalence of resistance to homologous antimicrobials, MDR, and AMU quantity are suggestive that the turkey sector's AMU strategy is achieving its desired impact. However, antimicrobials previously eliminated

**Funding:** The author(s) received no specific funding for this work.

**Competing interests:** The authors have declared that no competing interests exist.

for preventive use in turkey flocks and the use of highly important antimicrobials in human medicine suggest that the AMU reduction strategy should be monitored and re-evaluated periodically to mitigate the emergence of MDR bacteria and safeguard animal and public health.

## Introduction

The emergence of antimicrobial resistance (AMR) in enteric bacteria of humans and animals is a global health threat that has decreased treatment efficacy and increased the cost of infections with antimicrobial-resistant bacteria [1, 2]. It is projected that by 2050 an estimated 10 million AMR-related infections will occur worldwide [3]. Surveillance systems play a critical role in the mitigation and containment of AMR by monitoring antimicrobial use (AMU) and detecting emerging trends or unusual resistance phenotypes and/or genotypes in bacteria from the human-animal-environment interface [4]. Farm-level surveillance provides an opportunity for understanding turkey industry contextual details through the simultaneous collection of relevant demographics, risk factors, reasons for AMU, dose, and duration of treatment as well as AMR information from the same epidemiological unit and thus is more informative in the formulation of practical and effective flock health prevention strategies to curb AMR [5]. Recognizing that AMR has both food safety and public health implications, surveillance programs have been implemented in several countries long before the publication of the World Health Organization's Global Action Plan on AMR and the quadripartite' s global effort to address AMR [3, 6–13]. The National Antimicrobial Resistance Monitoring System (NARMS) in the USA, Danish Integrated Antimicrobial Resistance Monitoring and Research Programme (DANMAP) in Denmark, the European Antimicrobial Resistance Surveillance Network in Veterinary Medicine (EARS-Vet) in Europe and the Canadian Integrated Program for Antimicrobial Resistance Surveillance (CIPARS) in Canada are some examples of surveillance programs that include monitoring AMR in indicator, foodborne and/or pathogenic bacteria from turkeys [7, 8, 10–12, 14, 15]. Farm-level AMU is also collected via CIPARS turkey farm surveillance in Canada and various countries in the E.U. [16].

Canada is among the top ten turkey-producing countries globally [17], where approximately 500 registered commercial turkey farms produce over 155 million kg of turkey meat annually. The turkey sector contributed over $1.2 billion to Canada's Gross Domestic Product (GDP) [18, 19]. In 2021, Canada exported 8.3 million poults to 13 countries and 16.5 million kg of turkey meat to 40 countries [19]. From domestic consumption and international trade perspectives, surveillance has a substantive role in ensuring the microbial quality of live turkeys and their products.

Previous research studies in Canada characterized AMR in indicator and foodborne bacteria from turkey flocks and retail turkey meats and found antimicrobial-resistant bacteria and AMR phenotypes that have potential food safety implications [20–22]. Similarly, other turkey-producing countries have reported high levels of multidrug resistance (MDR) (resistance to $\geq 3$ antimicrobial classes) in *E. coli* isolates and resistance to WHO's Highest Priority Critically Important Antimicrobials (HP-CIA) including beta-lactams, 3rd, and 4th generation cephalosporins, aminopenicillins, fluoroquinolones and polymyxins (colistin) [23–27]. Studies have also shown that the use of antimicrobials in turkey flocks was significantly associated with the development of resistance to the same or unrelated antimicrobials [26–29]. To prevent the long-term implications of AMR on animal and public health, Canadian poultry producer groups implemented an AMU strategy [26, 29]. Monitoring AMU and AMR is one tool that

could track the progress of this strategy. In 2013, AMR surveillance in Canadian turkey was initiated in one province as part of the FoodNet Canada sentinel site, and the surveillance expanded to the 3 major turkey-producing Canadian provinces in 2016 [14]. The AMU strategy involves the progressive elimination of the preventive use of medically-important antimicrobials to slow down the development of AMR while reducing the total quantity of antimicrobials used in the sector. Step 1 of the strategy was introduced in May 2014 that eliminated the preventative use of $3^{rd}$ generation cephalosporins and fluoroquinolones. Step 2 was implemented at the end of 2018 and eliminated the preventive use of aminoglycosides, macrolides, penicillins, and streptogramins. Between 2019 and 2020, the incorporation of the Step 3 strategy ended the preventive use of bacitracin and tetracyclines [14, 26]. It is important to note that the strategy does not include the prohibition of the use of these antimicrobials for treatment of disease under veterinary supervision in the face of an outbreak. An initial descriptive analysis of the impact of these initiatives through CIPARS farm surveillance illustrated a rising proportion of isolates that exhibited susceptibility to all antimicrobials included in the NARMS AMR panel [14].

To better understand the impact of the stepwise AMU strategy on the prevalence of AMR and MDR in *E. coli* isolates of turkey flocks between 2016 and 2021, this study aimed to 1) estimate the prevalence of AMR, MDR and clustering patterns of resistance to antimicrobials across years, and 2) to investigate the temporal and geographical variations in AMR.

## Materials and methods

### Sentinel-farm surveillance

The CIPARS sentinel farm surveillance started in 2013 in one province of Canada, then gradually expanded to 5 turkey-producing provinces/regions in 2016 to harmonize with other commodities sampled at the farm level. Because of limited geographic coverage in the early phases of the program, this present study included farm data collected between 2016 and 2021.

The selection of turkey farms and flocks was based on the specific exclusion and inclusion criteria as described previously (e.g., compliant with the on-farm food safety program, not backyard or free-range turkeys, and farms drawn from diverse geographical areas and different farm production profiles) [14, 15]. Written consent was obtained from each selected turkey producer before sampling and data collection. Each veterinarian visited the selected farm during the last week of the turkey's grow-out period according to their targeted market weight [14, 15]. During each visit, veterinarians collected four pooled fecal samples (one from each of the four quadrants of the selected barn) per flock per farm per visit and submitted the samples according to CIPARS methodology for transport/shipment. The National Microbiology Laboratory, Public Health Agency of Canada, conducted bacterial culture and antimicrobial susceptibility testing. During each visit, veterinarians also administered a questionnaire to collect data on AMU, biosecurity, and relevant farm management practices.

### Bacterial culture and antimicrobial susceptibility testing

Bacterial culture and antimicrobial susceptibility testing methods are described in detail elsewhere [14, 15]. Briefly, 25 g of feces from each collected sample was mixed with 225 mL of buffered peptone water, then incubated at 35 ± 1˚C for 24 hours, and a drop from this incubated mixture was streaked onto a MacConkey agar plate before incubating the plate at 35 ˚C for 18–24 h. Suspected lactose-fermenting colonies were subsequently transferred onto Luria-Bertani agar and confirmed as *E. coli* using biochemical tests, Simmons citrate, and indole tests. Colonies negative on the indole tests were further confirmed using API$^{®}$ 20E bacterial identification kit.

A broth microdilution method was performed with the Sensititre Antimicrobial susceptibility testing (AST) system (Trek™ Diagnostic Systems Ltd, West Sussex, England) on one *E. coli* isolate per positive fecal sample. For *E. coli*, the CMV4AGNF designed by NARMS was run on the AST system to detect resistance to the following antimicrobials: amoxicillin-clavulanic acid, ceftriaxone, ciprofloxacin, ampicillin, azithromycin, cefoxitin, gentamicin, meropenem, nalidixic acid, streptomycin, trimethoprim-sulfamethoxazole, chloramphenicol, sulfisoxazole, and tetracycline. Beginning in 2019, the updated CMV5AGNF plate containing colistin was used. For each isolate, minimum inhibitory concentration (MIC) values were obtained. Based on the MIC breakpoints, these isolates were classified as susceptible, intermediate, or resistant following the Clinical and Laboratory Standards Institute (CLSI) M7-A8 guidelines. As CLSI interpretive criteria for Enterobacteriaceae were not available for azithromycin or streptomycin, breakpoints were determined according to the distribution of MIC values and were harmonized with those of the NARMS of the United States. *Escherichia coli* ATCC 25922 strains were used for quality control. Streptomycin was subsequently removed from the NARMS panel affecting 2020–2021 data; thus, interpretation for resistance to streptomycin was not available for those years.

## Data management

For this study, the intermediate isolates were classified as susceptible following both the NARMS (i.e., any isolate below the clinical resistance breakpoint using CLSI or NARMS/ CIPARS interpretative criteria) and the European Committee on Antimicrobial Susceptibility Testing (EUCAST) Steering Committee guidelines [30, 31]. Each antimicrobial susceptibility result of isolates determined as susceptible or resistant was converted into binomial outcome variables (susceptible as '0' and resistant as '1') for each drug. A multidrug-resistant (MDR) isolate was defined as an isolate that exhibited resistance to antimicrobials belonging to three or more antimicrobial classes [32]. The MDR variable had a binomial outcome (resistance $\geq 3$ antimicrobial classes as "1" and those $<3$ classes as "0").

Antimicrobial use practices on each turkey farm were collected through a questionnaire administered by veterinarians to turkey producers/staff. For this study, the weight-based indicator, milligrams per kg animal biomass (mg/kg turkey biomass) was used to measure antimicrobial quantity via all administration routes ($AMU_{anyroute}$) and was aggregated at the flock level for each antimicrobial class. The count-based indicator (binomial variables, yes or no) of AMU for each antimicrobial class at the flock level was determined and also used in subsequent analysis.

## Statistical analysis

Statistical analyses and data visualization were performed using the packages "lme4", "sjplot", "heatmap", "ggeffects", "ggplot" in R [33] software and R-studio (Version 1.4.1106© 2009–2021 RStudio, PBC) platform.

**Descriptive statistics.** The yearly surveillance level aggregate of AMU measured in mg/kg turkey biomass for each antimicrobial class and across all routes of administration ($AMU_{anyroute}$), and flock-level total AMU ($AMU_{total-any\ route}$) were determined. For this AMU measurement, descriptive statistics (mean, standard deviation, and range) were obtained. Count-based AMU outcomes (i.e., the proportion of flocks using a given antimicrobial) were also obtained.

Prevalence estimates for resistance to homologous antimicrobials and MDR per year and province were calculated using multivariable mixed-effect logistic regression models, accounting for flock-level clustering. The intercepts ($\beta_0$) from the null models were used to calculate

adjusted prevalence estimates (Pa) using the following formula [34]:

$$Pa = \frac{e^{\beta 0}}{1 + e^{\beta 0}}$$

Resistance to homologous antimicrobials and MDR were described following the European Food Safety Authority's methodology for describing levels of resistance [35],: rare: < 0.1%, very low: 0.1% to 1.0%, low: > 1.0% to 10.0%, moderate: > 10.0% to 20.0%, high: > 20.0% to 50.0%, very high: > 50.0% to 70.0% and extremely high: > 70.0%.

To assess the similarity of *E. coli* isolates in terms of their resistance to different antimicrobials for each sampling year, a hierarchical single-linkage clustering dendrogram (heatmap) was constructed, applying Ward's hierarchical clustering method with Euclidean distances [36]. Heatmaps were also constructed for the resistance patterns of *E. coli* isolates by province and year.

**Regression analyses.** Outcome variables with >5% or <95% prevalence were used for regression analysis. To account for clustering at the turkey-flock-level mixed-effect regression models were constructed by including turkey flocks as random intercepts. Mixed-effects multivariable Poisson regression models were built for the count outcome variable signified by the total antimicrobials used in turkey flocks ($AMU_{\text{total-any route}}$) to determine the changes in AMU across years and provinces. Overdispersion of the Poisson mixed effect model was assessed by using a goodness of fit chi-squared test, and if it was significant a negative mixed effect binomial model was used.

Mixed effect multivariable logistic regression models were built for each binomial outcome variable (resistance to individual antimicrobial classes and MDR) to evaluate the yearly changes in resistance and differences between provinces. Data from Alberta, British Columbia, and Saskatchewan were combined and defined as the Western region. As streptomycin was excluded in the 2020 and later NARMS panels, the streptomycin model utilized data only up to and including 2019. A likelihood ratio test was performed to evaluate the model's interaction between year and province/region.

## Results

### AMU in Canadian turkey farms (2016–2021)

The AMU information (S1 Table) was collected from 510 turkey farms, of which 332 farms were reported to have used medically important antimicrobials. The highest proportion of farms reported to have used antimicrobials was in 2016 (n = 60/72, 83.3%), followed by 2017 (n = 59/74, 79.7%), 2018 (n = 63/95, 66.3%), 2019 (n = 65/98, 66.3%), 2020 (n = 38/61, 62.3%), and 2021 (n = 47/110, 42.7%). The proportion of farms that used antimicrobials was highest in the Western region (n = 157/209), followed by Ontario (n = 100/153) and Québec (n = 75/148). The mg/kg turkey biomass by region is presented in S2 Table indicating lower AMU in Ontario and Québec in 2021 compared to the previous years. AMU by antimicrobial class from 2016 to 2021 is shown in Table 1 and Fig 1.

While AMU represented by mg/kg turkey biomass decreased for most of the antimicrobial classes in 2021, the following deviations were observed: a substantial rise in mean flock-level beta-lactam use in mg/kg turkey biomass, a re-emergence of the use of streptogramins (virginiamycin antimicrobial supposedly eliminated for preventive purposes at the end of 2018), an increase in tetracycline use in mg/kg turkey biomass, limited use of fluoroquinolones, and a slight drop in orthosomycins use in mg/kg turkey biomass.

A negative binomial mixed-effect regression analysis to determine the change of AMU between years and provinces also showed a significant reduction in the use of antimicrobials in each province in 2020 and 2021 (Fig 2).

**Table 1. Summary of antimicrobial use (AMU) by year, using count-based (proportion of flocks reported to have used a given antimicrobial) and quantity-based indicators (flock-level AMU in mg/kg turkey biomass).**

| Antimicrobials | YEAR | Count-based (n/N)[a] | Quantity based AMU (mg/kg biomass) at the flock level | |
|---|---|---|---|---|
| | | | Mean (SD)[b] | Range |
| Aminoglycosides | 2016 | 0.81 (58/72) | 0.26 (0.69) | 0–4.8 |
| | 2017 | 0.72 (53/74) | 0.11 (0.21) | 0–1.8 |
| | 2018 | 0.12 (11/95) | 0.11 (0.75) | 0–7.04 |
| | 2019 | 0.02 (2/98) | 0.26 (2.94) | 33.44 |
| | 2020 | 0 (0/61) | 0 (0) | 0–0 |
| | 2021 | 0.009 (1/110) | 0 (0) | 0–0.03 |
| Bacitracin | 2016 | 0.36 (26/72) | 23.36 (36.07) | 0–165.92 |
| | 2017 | 0.38 (28/74) | 24.57 (35.89) | 0–110.81 |
| | 2018 | 0.27 (26/95) | 18.91 (35.17) | 0–123.50 |
| | 2019 | 0.59 (58/98) | 37.23 (37.87) | 0–132.67 |
| | 2020 | 0.28 (17/61) | 11.84 (22.99) | 0–102.26 |
| | 2021 | 0.11 (12/110) | 6.07 (19.84) | 0–108.13 |
| Beta-lactams | 2016 | 0.15 (11/72) | 0.66 (2.23) | 0–12.57 |
| | 2017 | 0.09 (7/74) | 0.76 (3.54) | 0–23.31 |
| | 2018 | 0.14 (13/95) | 1.24 (4.68) | 0–34.7 |
| | 2019 | 0.08 (8/98) | 3.67 (18.57) | 0–137.25 |
| | 2020 | 0.11 (7/61) | 0.32 (2.41) | 0–18.09 |
| | 2021 | 0.09 (10/110) | 5.84 (26.65) | 0–227.59 |
| Folate Pathway Inhibitors | 2016 | 0.06 (4/72) | 1.87 (8.21) | 0–47.79 |
| | 2017 | 0.09 (7/74) | 7.36 (26.81) | 0–173.35 |
| | 2018 | 0.04 (4/95) | 4.02 (21.11) | 0–149.61 |
| | 2019 | 0.06 (6/98) | 4.05 (19.24) | 0–133.47 |
| | 2020 | 0.1 (6/61) | 5.68 (18.3) | 0–89.79 |
| | 2021 | 0.01 (1/110) | 0.48 (4.98) | 0–51.76 |
| Macrolides | 2016 | 0.07 (5/72) | 1.66 (7.53) | 0–41.09 |
| | 2017 | 0.05 (4/74) | 1 (5.12) | 0–30.57 |
| | 2018 | 0 (0/95) | 0 (0) | 0–0 |
| | 2019 | 0 (0/98) | 0 (0) | 0–0 |
| | 2020 | 0 (0/61) | 0 (0) | 0–0 |
| | 2021 | 0 (0/110) | 0 (0) | 0–0 |
| Orthosomycin | 2016 | 0 (0/72) | 0 (0) | 0–0 |
| | 2017 | 0 (0/74) | 0 (0) | 0–0 |
| | 2018 | 0.03 (3/95) | 0.42 (0.42) | 0–18.57 |
| | 2019 | 0.07 (7/98) | 0.81 (0.81) | 0–16.27 |
| | 2020 | 0.21 (13/61) | 3.35 (3.35) | 0–32.84 |
| | 2021 | 0.18 (20/110) | 3.09 (3.09) | 0–35.85 |
| Quinolones/Fluoroquinolones | 2016 | 0 (0/72) | 0 (0) | 0–0 |
| | 2017 | 0.01 (1/74) | 0.01 (0.05) | 0–0.42 |
| | 2018 | 0.01 (1/95) | 0.01 (0.05) | 0–0.49 |
| | 2019 | 0.02 (2/98) | 0.04 (0.28) | 0–2.35 |
| | 2020 | 0 (0/61) | 0 (0) | 0–0 |
| | 2021 | 0.01 (1/110) | 0 (0.03) | 0–0.28 |

(*Continued*)

**Table 1.** (Continued)

| Antimicrobials | YEAR | Count-based (n/N)[a] | Quantity based AMU (mg/kg biomass) at the flock level | |
|---|---|---|---|---|
| | | | Mean (SD)[b] | Range |
| Streptogramins | 2016 | 0.38 (27/72) | 9.86 (13.84) | 0–42.42 |
| | 2017 | 0.36 (27/74) | 8.29 (13.06) | 0–62.01 |
| | 2018 | 0.37 (35/95) | 9 (13.37) | 0–42.82 |
| | 2019 | 0.05 (5/98) | 0.6 (3.28) | 0–27.49 |
| | 2020 | 0 (0/61) | 0 (0) | 0–0 |
| | 2021 | 0.01 (1/110) | 0.33 (3.5) | 0–36.4 |
| Tetracyclines | 2016 | 0.08 (6/72) | 4.13 (16.51) | 0–71.65 |
| | 2017 | 0.03 (2/74) | 0.62 (4.87) | 0–41.42 |
| | 2018 | 0.07 (7/95) | 1.39 (9.37) | 0–84.19 |
| | 2019 | 0.05 (5/98) | 1.91 (11.29) | 0–108.99 |
| | 2020 | 0.05 (3/61) | 0.71 (4.58) | 0–30.86 |
| | 2021 | 0.04 (4/110) | 1.16 (7.39) | 0–58.80 |

[a] n = number of farms that used antimicrobials/N = total number of farms sampled

[b] average (SD = standard deviation)

## AMR in *E. coli* isolates

Between 2016 and 2020, 1976 *E. coli* isolates were obtained from turkeys in 503 out of 510 flocks surveyed. This study excluded seven flocks due to missing AMR data. One *E. coli* isolate was recovered from one flock, two isolates each from 6 flocks, three isolates each from 21 flocks, and four isolates each from the remaining 475 flocks. Antimicrobial susceptibility testing indicated that 69.0% (1363/1976) of isolates were resistant to at least one of the fourteen antimicrobials included in the CMV5AGNF NARMS Gram-negative panel. Of the 1363 resistant isolates, the most frequently occurring AMR pattern was resistance to one (19.1%) and two (19.2%) antimicrobials, whereas 1.0% (14/1363) of isolates were resistant to $\geq 7$ antimicrobials.

**The overall trend of AMR in *E. coli* from 2016 to 2021.** A summary of the AMR in *E. coli* isolates collected between 2016 and 2021 is presented in S3 Table. The six-year adjusted prevalence ($P_{\text{adj-total}}$) of tetracycline resistance in 1976 *E. coli* isolates was very high (59.4%). Similarly, high resistance to streptomycin (33.0%), ampicillin (25.8%), and sulfisoxazole (24.1%) were detected. Low to very low resistance was found for the remaining antimicrobials, including gentamicin (9.7%), nalidixic acid (1.6%), amoxicillin-clavulanic acid (1.5%), cefoxitin (1.3%), ceftriaxone (0.96%), ciprofloxacin (0.35%), azithromycin (0.1%), chloramphenicol (0.06%), and trimethoprim-sulfamethoxazole (0.01%). None of the isolates exhibited resistance to meropenem and colistin. There were year-wise variations in the number of *E. coli* isolates resistant to tetracyclines ($\chi^2 = 34.47$, p<0.000), sulfisoxazole ($\chi^2 = 25.16$, p<0.000), gentamicin ($\chi^2 = 53.53$, p<0.000), ampicillin ($\chi^2 = 14.22$, p = 0.01), and MDR ($\chi^2 = 77.61$, p<0.000). Geographical variations were observed in the distribution of the isolates resistant to tetracyclines ($\chi^2 = 6.8979$, p = 0.03), trimethoprim-sulfamethoxazole (69.29, p<0.000), streptomycin ($\chi^2 = 23.34$, p<0.000), gentamicin ($\chi^2 = 32.49$, p<0.000), and ampicillin ($\chi^2 = 17.92$, p = 0.0001).

**Temporal trends in AMR in *E. coli* isolates.** The adjusted prevalence ($P_{\text{adj}}$) of *E. coli* isolates resistant to individual antimicrobials between 2016 and 2021 is presented in Fig 3.

The $P_{\text{adj}}$ of tetracycline-resistant *E. coli* isolates decreased from extremely high resistance in 2016 to very high levels between 2017–2020, then declined further to a high level in 2021.

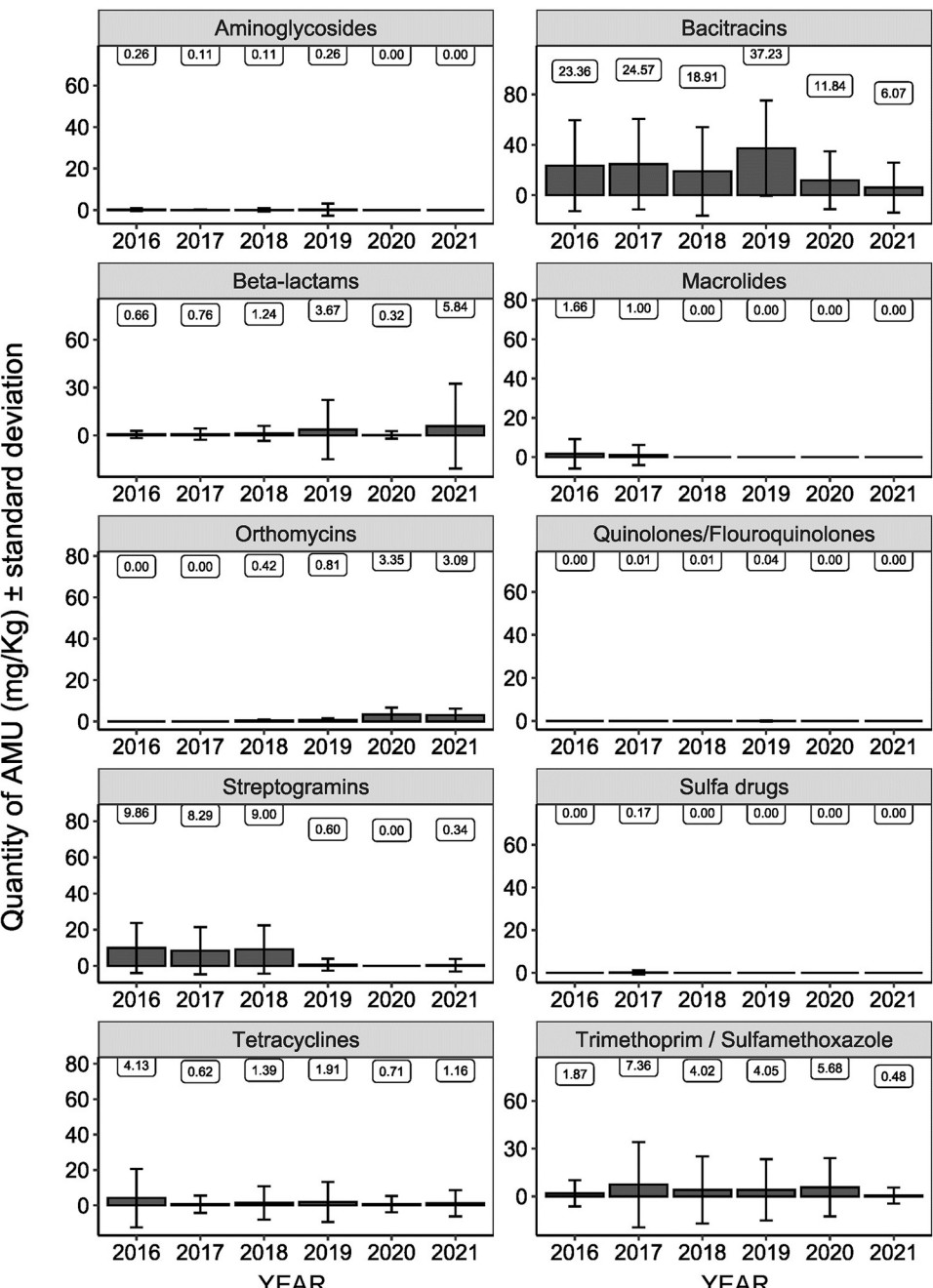

**Fig 1. Quantity-based antimicrobial use (AMU) indicators (flock-level AMU in mg/kg turkey biomass) by year.**

Whereas very high streptomycin-resistant *E. coli* isolates were found in 2017 (51.1%) but were lower in 2016, 2018, and 2019. Ampicillin resistance fluctuated over time, while the $P_{adj}$ of sulfisoxazole-resistant isolates was reduced from high resistance in 2016–2020 (21.3% - 32%) to a moderate level in 2021(16.6%). Similarly, low-level gentamicin resistance was observed in later years (2019–2021:8.2%-3.1%) compared to moderate levels in earlier surveillance years (2016–2018:19.2%-10.6%). Although the overall $P_{adj}$ for chloramphenicol-resistant isolates was very low, occurrences of chloramphenicol-resistant isolates were further reduced in 2021(0.04%) compared to previous years (0.06%-0.09%).

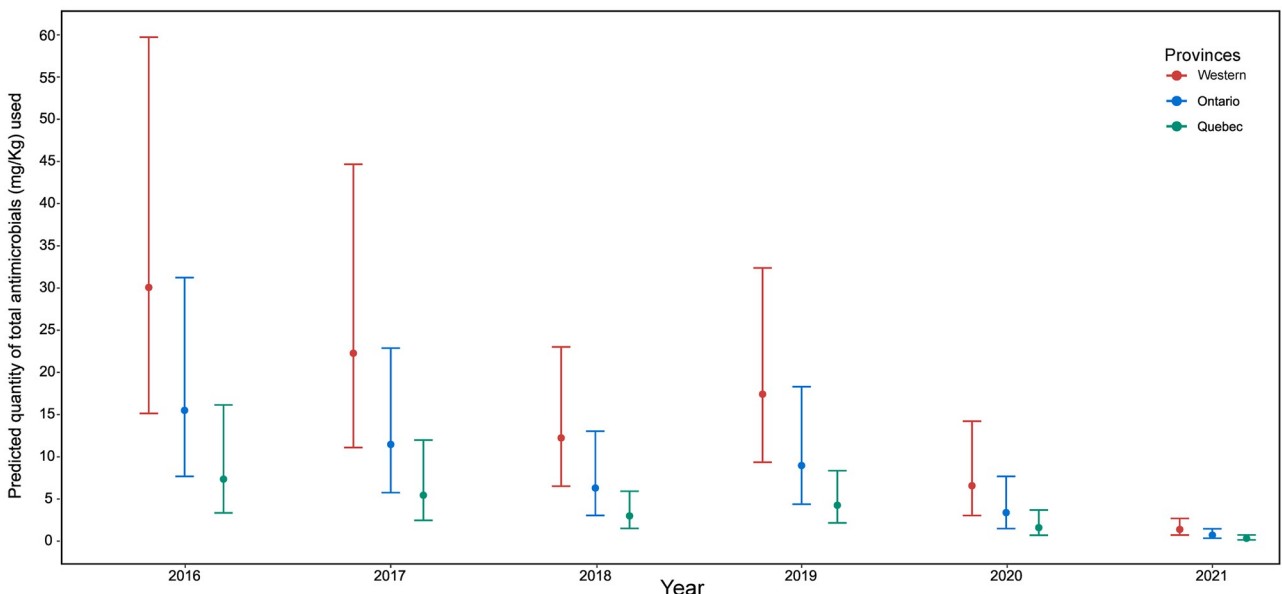

**Fig 2. Predicted total AMU$_{anyroute}$ (mg/kg) per year and province calculated from the mixed effect negative binomial regression analysis.** The western provinces include Alberta (n = 60 flocks), British Columbia (n = 147 flocks), and Saskatchewan (n = 2 flocks). The round dot denotes the predicted IRR of AMU for each province per year, and lines with error bars are 95% confidence intervals.

**AMR trends in *E. coli* isolates by provinces/region.** When data were compared between provinces/regions (Fig 4), *E. coli* isolates from all geographical locations yielded very high tetracycline resistance, the highest being in Ontario (64.2%).

Variations in resistance levels were observed in other antimicrobials. Isolates from Ontario exhibited high resistance to streptomycin and sulfisoxazole, moderate resistance to ampicillin, low resistance to gentamicin, and very low resistance to trimethoprim-sulfamethoxazole and chloramphenicol. The *E. coli* isolates from Québec exhibited high resistance to ampicillin, streptomycin, and sulfisoxazole, low resistance to gentamicin and trimethoprim-sulfamethoxazole, and very low resistance to chloramphenicol. In contrast, isolates from the Western region showed very high resistance to streptomycin, high resistance to sulfisoxazole and ampicillin, moderate resistance to gentamicin, and very low resistance to chloramphenicol and trimethoprim-sulfamethoxazole. Ampicillin-resistant (32.1%) and sulfisoxazole-resistant (27.7%) isolates were observed more frequently in Québec, whereas streptomycin-resistant (50.6%) and gentamicin-resistant isolates (16.0%) were observed more frequently in the Western region.

**Trends of MDR in *E. coli* isolates.** Overall, MDR was detected in 25.7% (95% CI: 22.7–29.0%) of the isolates over the six-year surveillance timeframe included in this study. The prevalence of MDR fluctuated over time, and a reduction was observed during the latter two years of the surveillance timeframe, specifically, one-year post Step 2 and 2-years post Step 3 of the turkey sector's AMU strategy. Adjusted prevalence of MDR was 38.2% (95% CI: 30.9–46.0%) in 2016, 41.0% (95% CI: 32.2–50.4%) in 2017, 27.4% (95% CI: 20.7–35.3%) in 2018, 29.1% (95% CI: 22.5–36.8%) in 2019, 19.5% (95% CI: 12.4–29.2%) in 2020, and 11.9% (95% CI: 7.8–17.5%) in 2021. The Western region had the highest adjusted prevalence of MDR (30.0%, 95% CI: 25.7–34.7%) followed by Québec (24.4%, 95% CI: 18.0–32.0%) and Ontario (21.4%, 95% CI:16.5–27.2%).

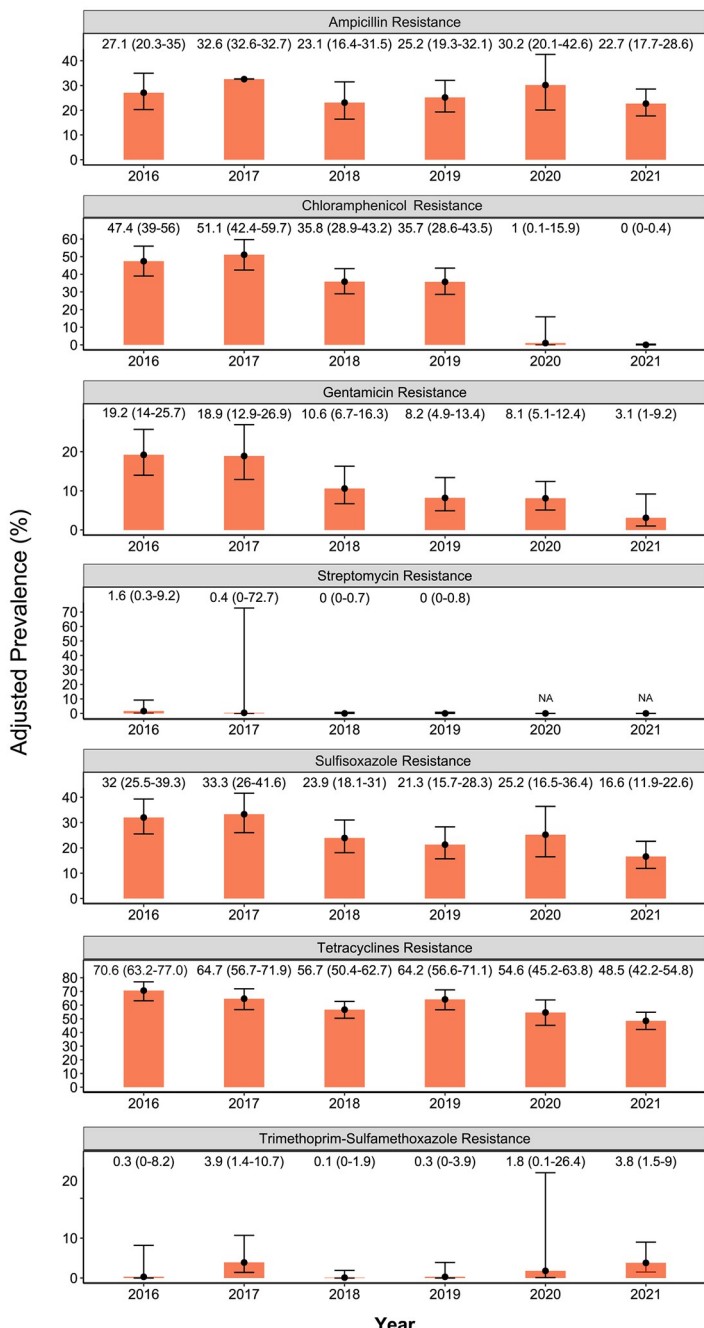

**Fig 3. Adjusted prevalence (y-axes) estimates of resistance to antimicrobials (x-axes) in *E. coli* isolates (n = 1976) of turkeys by year, accounting for turkey farm level clustering.** The lines with error bars in each orange bar are 95% confidence intervals (number within brackets in the figure) of the estimated prevalence. NA: Streptomycin resistance was not included on the testing panel in 2020 and 2021.

**Clustering analysis for resistance patterns in *E. coli* isolates.** Hierarchical clustering of the presence or absence of AMR in each isolate per year is displayed as a heatmap in Fig 5.

The column dendrogram of the heatmaps for the years 2016–2021 showed two major clusters: a cluster of resistance to tetracyclines and streptomycin for all years except in 2020 where

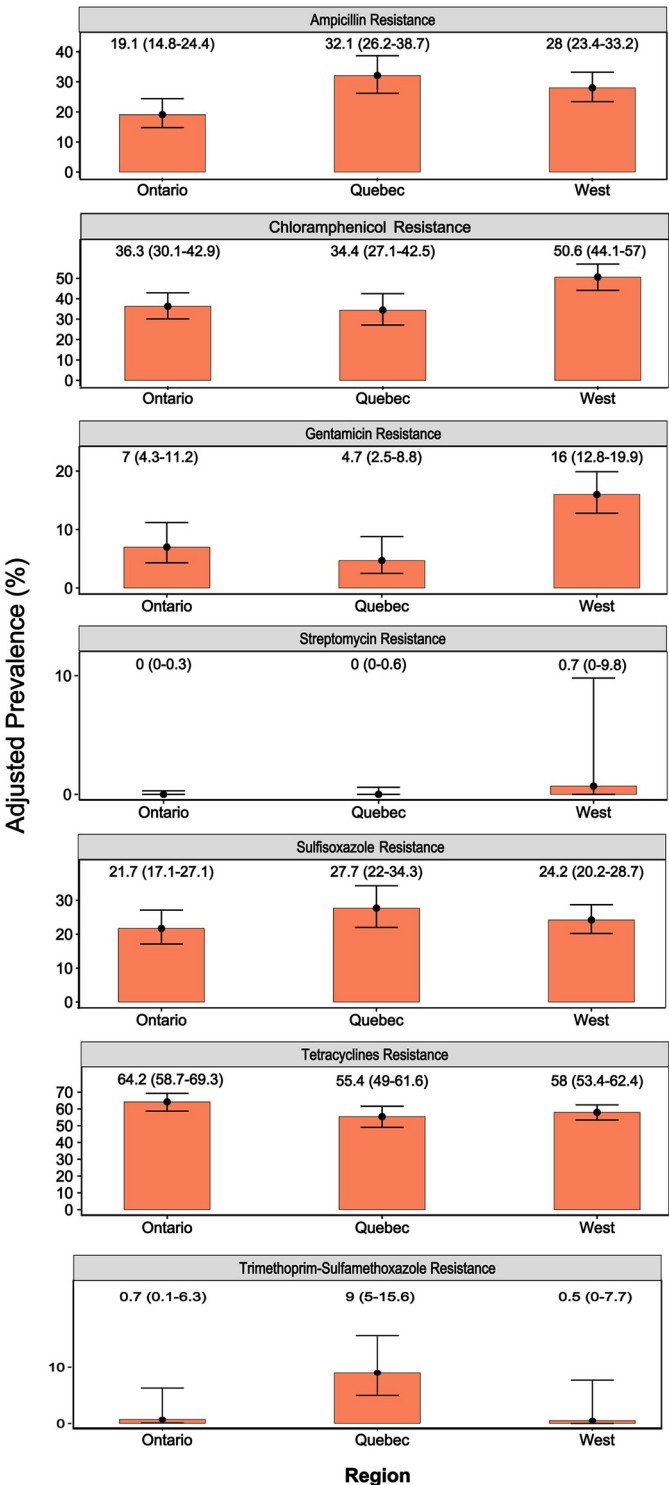

**Fig 4. Adjusted prevalence (y-axes) estimates of resistance to antimicrobials (x-axes) in *E. coli* isolates (n = 1976) of turkeys in each province, accounting for turkey farm level clustering.** The lines with error bars in each orange bar are 95% confidence intervals (number within brackets in the figure) of the estimated prevalence. The Western provinces include Alberta, British Columbia, and Saskatchewan.

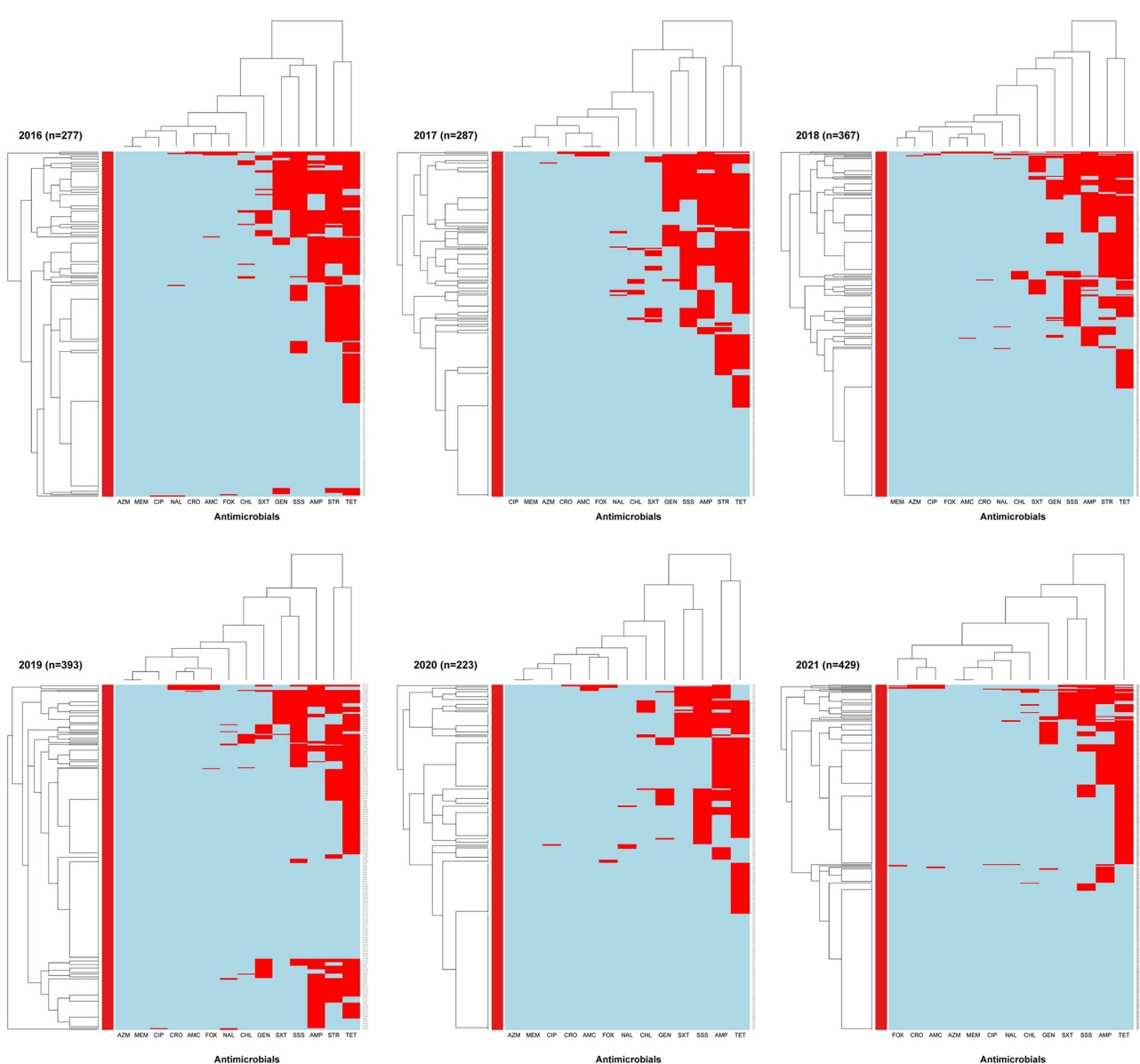

**Fig 5. Heatmap showing the hierarchal clustering of resistance to 14 antimicrobials in *E. coli* isolated from Canadian turkey flocks between 2016 and 2021.** X-axes represent the antimicrobial classes: amoxicillin (AMC), ampicillin (AMP), azithromycin (AZM), chloramphenicol (CHL), ciprofloxacin (CIP), ceftriaxone (CRO), cefoxitin (FOX), gentamicin (GEN), meropenem (MEM), nalidixic acid (NAL), sulfisoxazole (SSS), streptomycin (STR), trimethoprim-sulfamethoxazole (SXT), tetracyclines (TET). Y-axes represent the *E. coli* isolates included in this study. The blue color depicts susceptibility, and the red color illustrates resistant patterns.

tetracycline and ampicillin resistance clustered, and in 2021 where only tetracycline resistance clustered. The second AMR cluster contained >2 to 8 subclusters that varied significantly over the years. From 2016 to 2021, heat maps (rows) showed a decreased frequency of resistance and MDR *E. coli* isolates. In 2020 and 2021, the AMR pattern of tetracycline-trimethoprim-sulfamethoxazole-ampicillin-sulfisoxazole-gentamicin, whereas in 2016–2019 the AMR pattern of tetracycline-ampicillin-sulfisoxazole-trimethoprim-sulfamethoxazole-gentamicin was detected.

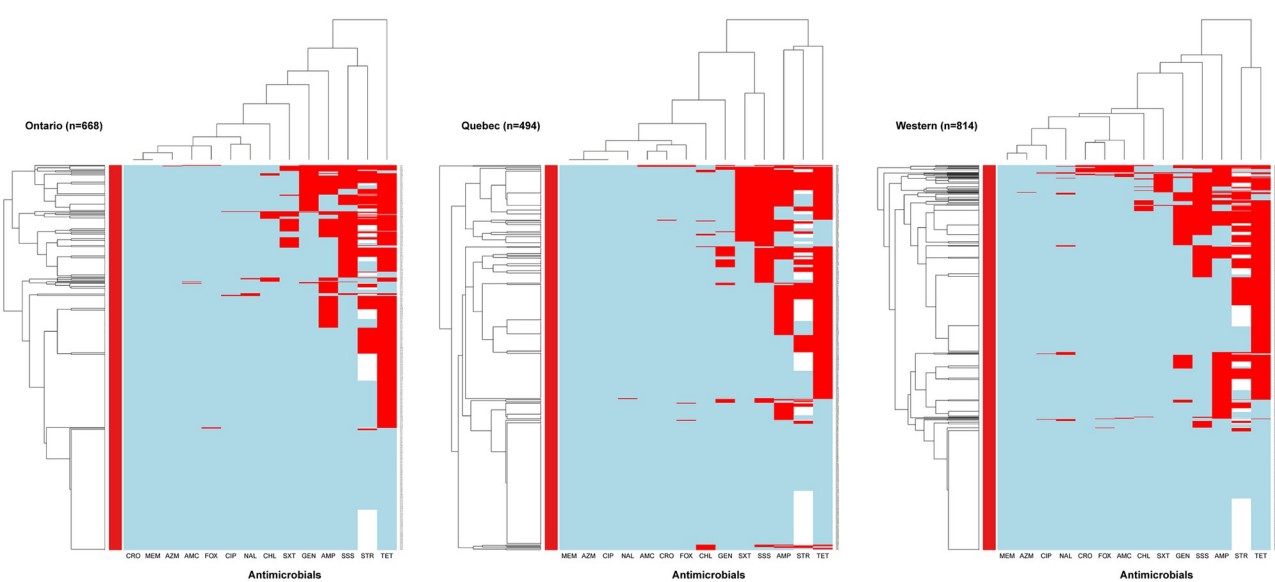

**Fig 6. Heatmap showing the hierarchical clustering of resistance to 14 antimicrobials in *E. coli* (n = 1986) isolated from Canadian turkey flocks by regions.** X-axes represent the antimicrobial agents: amoxicillin (AMC), ampicillin (AMP), azithromycin (AZM), chloramphenicol (CHL), ciprofloxacin (CIP), ceftriaxone (CRO), cefoxitin (FOX), gentamicin (GEN), meropenem (MEM), nalidixic acid (NAL), sulfisoxazole (SSS), streptomycin (STR), trimethoprim-sulfamethoxazole (SXT), tetracyclines (TET). Y-axes represent the *E. coli* isolates included in this study. The blue color depicts susceptibility, and the red color illustrates resistant patterns.

The row dendrograms of heatmaps (Fig 5) depict two distinct clusters of *E. coli* isolates. Susceptible *E. coli* isolates and isolates resistant to tetracycline in 2016 to 2018 and 2020, with sulfisoxazole and trimethoprim-sulfamethoxazole and chloramphenicol-resistant isolates from 2019–2021.

The differences in AMR patterns in *E. coli* isolates among the regions/provinces (Fig 6) found mostly similar AMR patterns in the column dendrograms, showing a cluster of resistance to tetracycline-streptomycin-ampicillin in Quebec and Western provinces, and a cluster of resistance to tetracycline-streptomycin-sulfisoxazole in Ontario. In the row dendrograms, the main *E. coli* isolate cluster was susceptibility to all antimicrobials in all regions/provinces, and the second cluster included isolates resistant to 2 to 8 different antimicrobials.

Further exploration of resistance patterns in *E. coli* isolates by regions and years (S1–S3 Figs) showed the reduction of AMR in 2020 and 2021 in all regions/provinces among *E. coli* isolates of turkey flocks.

**Regression analysis.** Table 2 and S4 Fig depicts the results of multivariable mixed-effect logistic regression models evaluating associations between resistance to individual antimicrobials of *E. coli* isolates and regions/provinces and years accounting for clustering at the turkey flock level.

The effect of the AMU reduction strategy, which removed the preventive use of certain antimicrobials can be explored by assessing the odds of AMR for each antimicrobial. The models indicated a significant reduction of gentamicin and tetracycline resistance in *E. coli* isolates in 2020 and 2021 compared to 2016, coinciding with the implementation of the turkey sector's AMU strategy as the likelihood of *E. coli* being resistant to gentamicin and tetracyclines was significantly lower after implementing Steps 2 and 3. Moreover, resistance to gentamicin in *E. coli* isolates has significantly decreased since 2018. A similar association was also observed for streptomycin resistance in 2018 and 2019 after the Step 2 strategy implementation. In contrast,

**Table 2. Associations between antimicrobial resistance to individual antimicrobials in *E. coli* isolates (n = 1986) and sampling years and regions determined by the multivariable mixed effect logistic regression analysis, accounting for clustering at turkey flock level.**

| Antimicrobial Resistance | Predictor variable | | Odds Ratio (95% CI[a]) | p-value |
|---|---|---|---|---|
| Ampicillin | Province/region | Western | Referent | |
| | | Ontario | 0.61 (0.43–0.87) | 0.01* |
| | | Quebec | 1.22 (0.85–1.76) | 0.29 |
| | Year | 2016 | Referent | |
| | | 2017 | 1.5 (0.87–2.59) | 0.14 |
| | | 2018 | 0.9 (0.53–1.51) | 0.68 |
| | | 2019 | 0.88 (0.52–1.47) | 0.61 |
| | | 2020 | 1.26 (0.7–2.25) | 0.44 |
| | | 2021 | 0.72 (0.43–1.21) | 0.21 |
| | Intercept | | 0.4 (0.26–0.62) | <0.001* |
| Gentamicin | Province/region | Western | Referent | |
| | | Ontario | 0.53 (0.37–0.76) | <0.001* |
| | | Quebec | 0.34 (0.21–0.53) | <0.001* |
| | Year | 2016 | Referent | |
| | | 2017 | 1.31 (0.78–2.21) | 0.3 |
| | | 2018 | 0.58 (0.34–0.99) | 0.05* |
| | | 2019 | 0.45 (0.26–0.77) | <0.001* |
| | | 2020 | 0.32 (0.16–0.62) | <0.001* |
| | | 2021 | 0.33 (0.19–0.58) | <0.001* |
| | Intercept | | 0.31 (0.21–0.48) | <0.001* |
| Sulfisoxazole | Province/region | Western | Referent | |
| | | Ontario | 0.92 (0.66–1.28) | 0.62 |
| | | Quebec | 1.32 (0.93–1.88) | 0.12 |
| | Year | 2016 | Referent | |
| | | 2017 | 1.12 (0.68–1.86) | 0.65 |
| | | 2018 | 0.7 (0.43–1.14) | 0.15 |
| | | 2019 | 0.61 (0.38–1) | 0.05 |
| | | 2020 | 0.82 (0.48–1.42) | 0.49 |
| | | 2021 | 0.45 (0.28–0.73) | <0.001* |
| | Intercept | | 0.43 (0.29–0.64) | <0.001* |
| Streptomycin[b] | Province/region | Western | Referent | |
| | | Ontario | 0.52 (0.36–0.76) | <0.001* |
| | | Quebec | 0.53 (0.35–0.8) | <0.001* |
| | Year | 2016 | Referent | |
| | | 2017 | 1.19 (0.73–1.95) | 0.48 |
| | | 2018 | 0.61 (0.38–0.97) | 0.04* |
| | | 2019 | 0.63 (0.39–1) | 0.05* |
| | Intercept | | 1.31 (0.88–1.96) | 0.18 |

(*Continued*)

**Table 2.** (Continued)

| Antimicrobial Resistance | Predictor variable | | Odds Ratio (95% CI[a]) | p-value |
|---|---|---|---|---|
| Trimethoprim / Sulfamethoxazole | Province/region | Western | Referent | |
| | | Ontario | 1.93 (0.95–3.93) | 0.07 |
| | | Quebec | 7.43 (3.51–15.72) | <0.001* |
| | Year | 2016 | Referent | |
| | | 2017 | 1.07 (0.35–3.26) | 0.91 |
| | | 2018 | 1.11 (0.38–3.21) | 0.85 |
| | | 2019 | 1.03 (0.36–2.93) | 0.96 |
| | | 2020 | 1.83 (0.58–5.82) | 0.31 |
| | | 2021 | 1.1 (0.4–3.07) | 0.85 |
| | Intercept | | 0.01 (0–0.03) | <0.001* |
| Tetracyclines | Province/region | West | Referent | |
| | | Ontario | 1.24 (0.93–1.64) | 0.14 |
| | | Quebec | 0.92 (0.67–1.25) | 0.58 |
| | Year | 2016 | Referent | |
| | | 2017 | 0.74 (0.47–1.18) | 0.20 |
| | | 2018 | 0.55 (0.36–0.85) | 0.01* |
| | | 2019 | 0.72 (0.47–1.11) | 0.14 |
| | | 2020 | 0.51 (0.31–0.83) | 0.01* |
| | | 2021 | 0.39 (0.26–0.6) | <0.001* |
| | Intercept | | 2.3 (1.59–3.31) | <0.001* |

*Significant at p <0.05;

[a]CI: Confidence interval;

[b]Not tested in 2020 and 2021.

these isolates had significantly lower resistance to sulfisoxazole in 2021 than in 2016, with inconsistent results in other years. Though not statistically significant, the resistance to tri-methoprim-sulfamethoxazole and ampicillin in 2021 was lower than in 2016.

Compared to the Western region, *E. coli* isolates from Ontario had significantly lower odds of being streptomycin resistant, whereas resistance to trimethoprim-sulfamethoxazole, ampicillin, and gentamicin was significantly higher in Ontario than in the Western region. Similarly, isolates from Québec had significantly higher odds of resistance to gentamicin and trimethoprim-sulfamethoxazole and lower odds of resistance to streptomycin. Though not statistically significant, the odds of tetracycline resistance were higher in Ontario than in the Western region, and lower in Quebec.

The predicted probabilities of resistance to each antimicrobial in *E. coli* isolates calculated from the mixed effect logistic regression analysis are shown in Fig 7.

Interestingly, MDR (Fig 8) in *E. coli* isolates was significantly lower in 2020 and 2021 compared to 2016, and this reduction in MDR was significantly observed in Ontario compared to the Western region.

## Discussion

In this study, we examined the AMU and AMR surveillance data over six years (2016–2021) collected by CIPARS from turkey farms across three major turkey-producing Canadian provinces/regions to evaluate the impact of a stepwise AMU reduction strategy on the prevalence of AMR in *E. coli* isolated from turkey flocks. The implementation of the AMU reduction

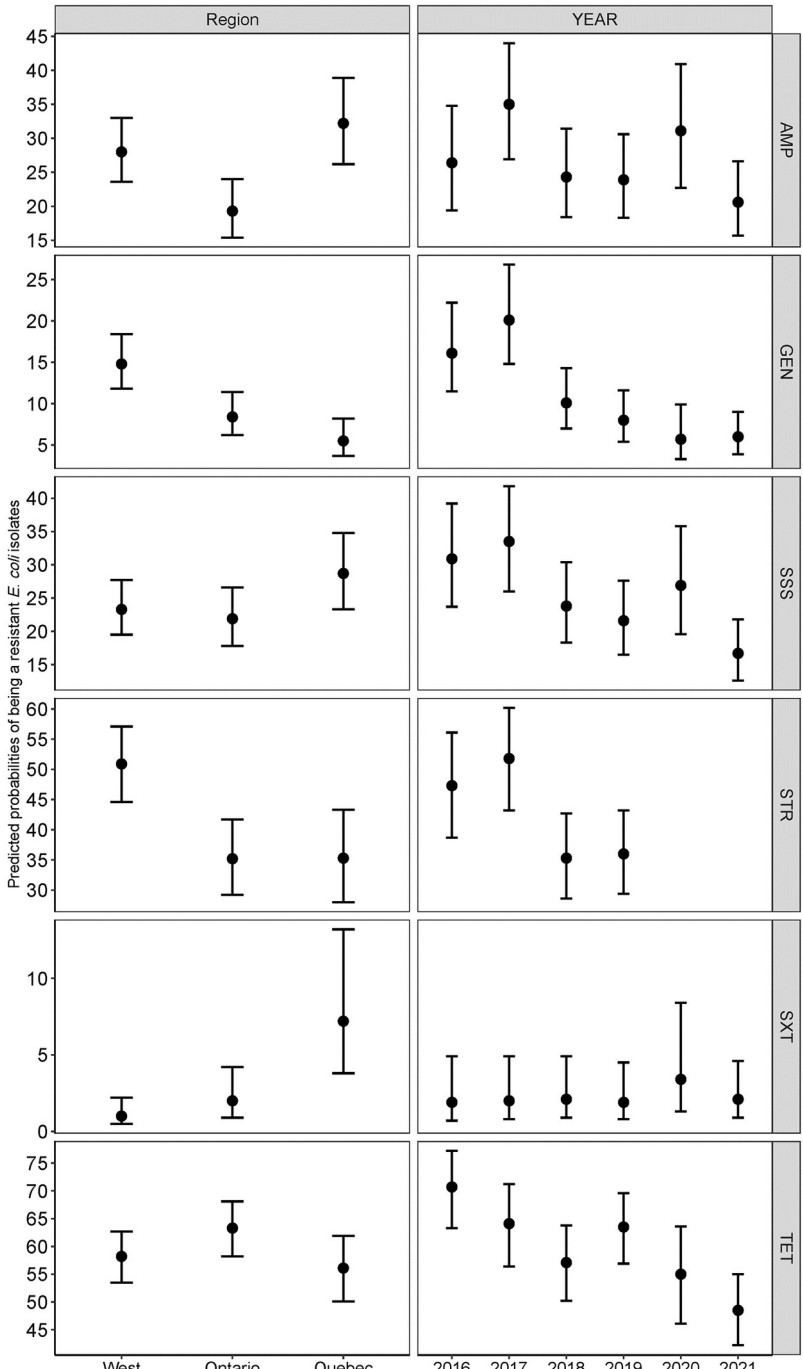

**Fig 7. Predicted probabilities of the *E. coli* isolate being resistant to each drug calculated from the multivariable mixed effect logistic regression analysis.**

strategy is one approach to promoting the stewardship of AMU. The poultry industry-wide AMU strategy had a high impact on reducing the prevalence of resistance to homologous and multiple antimicrobials in *E. coli* isolates from turkeys. In addition to the poultry industry-driven AMU reduction strategy, and the regulatory changes in dispensing veterinary drugs,

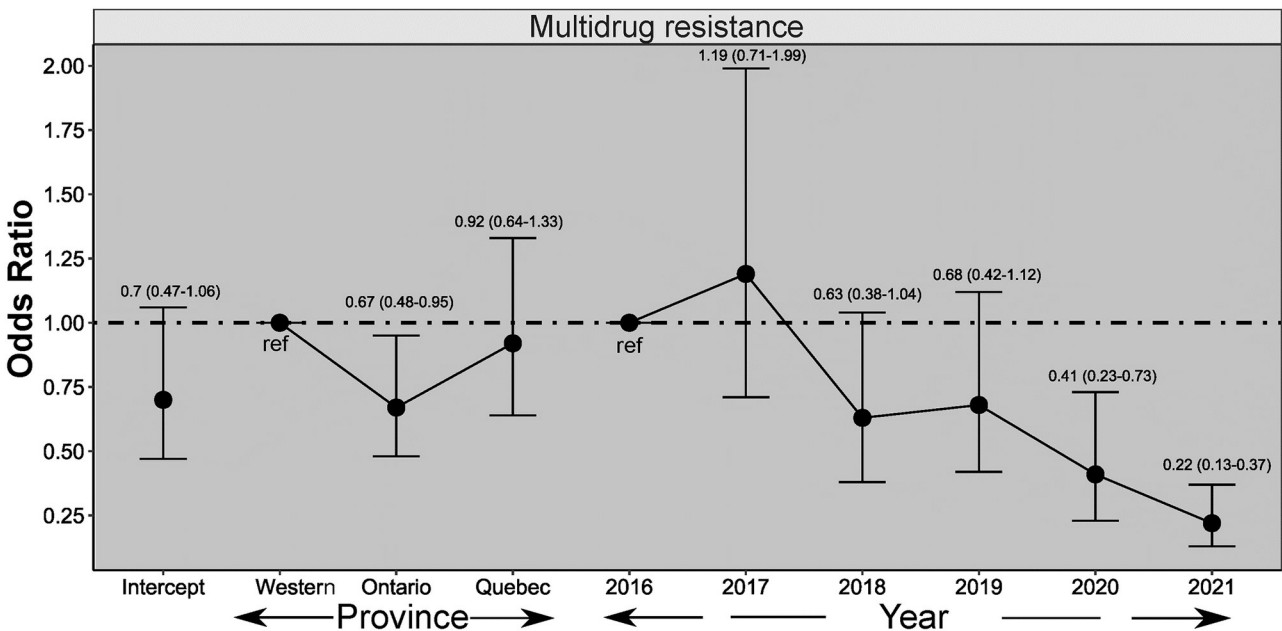

**Fig 8. Multivariable mixed effect logistic regression analysis results of associations among multidrug resistance in *E. coli* isolates (n = 1986) of Canadian turkey flocks and year and region.**

implementing the *"Regulations Amending the Food and Drug Regulations–Veterinary Drugs–Antimicrobial Resistance"* [37], have also contributed to enhancing veterinary oversight and AMU stewardship. Antimicrobial stewardship has shown to be effective in reducing the emergence of AMR in enteric bacteria [38] and is emphasized in AMR action plans/frameworks/strategies at various levels (international, national, and local industry) to reduce the spread of AMR bacteria in the human-animal-environment interface. The voluntary involvement and continuous support of Canadian turkey stakeholders in AMU and AMR surveillance and implementation of an antimicrobial stewardship program helped to mitigate the emergence of AMR in *E. coli* of turkey flocks. These changes are also expected to contribute to food safety and ultimately reduce the burden of infections with multidrug-resistant bacteria in animals and humans.

Unlike in broiler chickens, turkey producers rarely used 3<sup>rd</sup> generation cephalosporins which was reflected in the low-level resistance observed to these antimicrobials (ceftriaxone and cefoxitin) in this study. The preventive use of aminoglycosides, macrolides, penicillins, and streptogramins was eliminated at the end of 2018 at Step 2 of the AMU reduction strategy; however, the occasional use of certain antimicrobial classes for treatment purposes (not part of the strategy) was still observed in 2019–2021. Sporadic use of bacitracin in 2020–2021 and one flock using virginiamycin in 2021 was detected, even though their preventive use was eliminated at the end of 2019 and 2018, respectively. The use of these antimicrobials on a few farms (1–17) could be due to the necessity to treat bacterial infections in disease outbreak situations. The use of these antimicrobials was observed more frequently in the Western region and Québec and rarely in Ontario. Other surveillance information (e.g., disease diagnosis and farm-level risk factors) would be necessary to understand these provincial/regional variations in AMU practices.

In our study, *E. coli* isolates had very high resistance to tetracyclines, which is consistent with the previous AMR studies involving the turkey flocks in Canada [23, 39], European

countries [27, 40], and Egypt [41]. The presence of tetracycline resistance in *E. coli* in poultry is well-known as this is one of the older antimicrobials that has been used historically and is still used currently for disease prevention and treatment, which could explain the persistence of resistance to this antimicrobial. Our previous study showed that the use of tetracycline was positively associated with the increase in tetracycline-resistant *E. coli* isolates from turkey flocks [42]. Despite the reported use of tetracyclines, a significant reduction from high to moderate resistance to tetracyclines among *E. coli* isolates from 2016 to 2021 was noted. While the resistance levels have dropped, the ongoing concern is that the plasmid-encoded *tet* genes in *E. coli*, a determinant of tetracycline resistance, could transfer horizontally to other non-resistant bacteria in the gut or the animal environment leading to the maintenance of tetracycline resistance in animal populations [43]. Therefore, similar to 2021, in the future, we anticipate a continuous reduction of tetracycline resistance in commensal or foodborne bacteria from poultry, concomitantly with the further reduction of tetracycline use (including tetracycline combination products) and other antimicrobials such as streptogramins, known to co-select for tetracycline resistance [42].

The proportion of sulfisoxazole-resistant *E. coli* isolates also declined from high in 2016 to moderate in 2021, which is a favorable finding for the turkey industry as the emergence of antimicrobial-resistant bacteria could limit treatment options. Reduction of the use of this antimicrobial could also mean that the efficacy of folate pathway inhibitors is being preserved, and more importantly, the emergence of resistant bacteria and their potential to enter the food chain could be reduced [44]. In human medicine, sulfisoxazole alone or in combination with trimethoprim is one of the inexpensive second and third-line drug choices to treat acute bacterial meningitis, bacterial infections of the respiratory tract, or non-typhoidal *Salmonella* infections [45]. In 1989, a waterborne outbreak was associated with sulfisoxazole-resistant *E. coli* O157:H7 in Missouri, USA [46]. Therefore, reduced resistance to sulfisoxazole has public health benefits. However, our study did not find much use of sulfonamide antimicrobials alone from 2018–2021, but the folate pathway inhibitor antimicrobial combination, trimethoprim-sulfadiazine, was reported to have been used in turkey flocks for the treatment of respiratory infections and septicemia, a sequela of colibacillosis. The persistence of sulfisoxazole resistance is likely due to the co-selection for resistance, as studies have shown the linkages of multiple resistance genes on the same plasmids or transposons resulting in resistance to the same or dissimilar antimicrobials [47–49]. A small study conducted in Canadian poultry flocks also showed that either tetracyclines or aminoglycosides (kanamycin or streptomycin) use could enhance the resistance to sulfonamides in *E. coli*, which aligns with our study's occasional use of tetracyclines and aminoglycosides [50]. Heatmaps in this study also indicated the clustering of resistance to tetracyclines with sulfisoxazole and ampicillin. Additionally, strong associations were reported between the *tet*A and *sul*1 genes that carry the phenotypic resistance to tetracyclines and sulfonamides, respectively [43]. Molecular epidemiological studies assessing the role of AMR genes in the persistence, selection, and transmission of AMR determinants in *E. coli* isolates of turkey flocks would provide a better understanding of the long-term implications of using trimethoprim and sulfonamides.

Beta-lactams are used to treat or prevent several bacterial infections in poultry. They are also one of the critically important antimicrobials for human medicine based on the Health Canada Veterinary Drugs Directorate and World Health Organization categorization system [6, 51, 52]. In this study, resistance to ampicillin varied during the study period. Though not statistically significant, we observed a decrease in the ampicillin-resistant *E. coli* isolates in turkey flocks in 2021. It has been shown that β-lactamases, particularly $bla_{TEM}$ variants and amp genes, are responsible for ampicillin resistance in *E. coli* [53]. Despite the absence of certain antibiotic usage in the poultry industry, ampicillin resistance has been detected in

some studies [54]. Turkey farms also did not use penicillin extensively in our study, except in a few flocks (<13 of 510 farms), yet we observed a high proportion of resistance to ampicillin (22–31%). As described earlier and in our previous study [40], the emergence of resistance to a particular antimicrobial is possible due to the use of the same or different antimicrobials (cross-selection) or via vertical transmission of the resistance genes among *E. coli* isolates. Therefore, further identification of the genetic and farm-level determinants of AMR could contribute to our understanding of the persistence of beta-lactamase-resistant *E. coli* isolates in turkey flocks.

The elimination of the preventive use of aminoglycosides (in particular gentamicin, reported to have been used at the hatchery for treating colibacillosis) was implemented at the end of 2018 in the Canadian turkey flocks. Consequently, we observed a significant drop in the frequency of gentamicin-resistant *E. coli* isolates in the following years. Rare to low-level resistance was observed in other antimicrobials deemed as HP-CIA's by the WHO (3rd generation cephalosporins, macrolides, and quinolones) [6, 52].

It is well-recognized that AMU exerts selection pressure on commensal and pathogenic *E. coli* isolates. Consequently, these bacteria persist in the environment and, over time, have the potential to emerge to become multidrug resistant, which can serve as a reservoir for mobile genetic elements [55]. Despite MDR *E. coli* in Canadian turkey flocks being uncommon, the turkey industry's AMU reduction strategy has resulted in a further reduction in the prevalence of MDR *E. coli* isolates. This is a positive finding as the reduction in multidrug-resistant isolates is one of the desired outcomes of the turkey sector's AMU strategy. Ampicillin-streptomycin-tetracyclines and streptomycin-sulfisoxazole-tetracyclines were two major MDR patterns found in this study and supported the previous theory of the presence of genes encoding these phenotypic resistances in the same plasmid or transposons [47–49].

This study also identified provincial/regional differences in the resistance patterns in *E. coli*, which could reflect regional variations in AMU practices that warrant investigation of the underlying data (i.e., other risk factors). Additionally, creating the heatmaps through clustering techniques proved to be a useful tool in assessing AMR patterns, in addition to routine surveillance and reporting of AMU and AMR for tracking the progress of the AMU strategy.

This current study examined AMU and AMR in one commensal organism and thus, results should be interpreted with caution. The stepwise antimicrobial reduction/elimination strategy was important in the preservation of the efficacy of the relevant antimicrobials and more importantly, for the containment of AMR, reducing its transmission downstream of the farm-to-fork continuum. As previously described, the antimicrobials targeted by the reduction strategy could still be used for treatment purposes (i.e., during outbreak situations under veterinary supervision), therefore, AMU reduction may not always correspond to an observed reduction in resistance. Simultaneous assessment of other surveillance components (including the full spectrum of the bacterial organism under surveillance) using a One Health approach and evaluation of the animal welfare and economic benefits of the AMU reduction required to understand the implications of the industry-led strategy.

## Conclusions

Reduction in AMU and AMR in Canadian turkey flocks was achieved, which is the intended goal of the turkey AMU strategy. The regional variations in the AMR patterns warrant a locally-focused stewardship program that addresses the needs of different production settings and disease pressures. Additionally, to limit the use of critically important antimicrobials to treat bacterial infections, an ongoing exploration of antimicrobial alternatives is needed to safeguard the sustainability of turkey production in Canada.

## Supporting information

**S1 Fig. A heatmap of antimicrobial resistance in *E. coli* isolates collected from Canadian turkey flocks by region in 2016 (n = 277) and 2017 (n = 287).** X-axes represent the antimicrobial classes: amoxicillin (AMC), ampicillin (AMP), azithromycin (AZM), chloramphenicol (CHL), ciprofloxacin (CIP), ceftriaxone (CRO), cefoxitin (FOX), gentamicin (GEN), meropenem (MEM), nalidixic acid (NAL), Sulfisoxazole (SSS), Streptomycin (STR), Trimethoprim-Sulfamethoxazole (SXT), Tetracyclines (TET). Y-axes represent the E. coli isolates included in this study. The blue color depicts susceptibility, and the red color illustrates resistant patterns.
(TIF)

**S2 Fig. A heatmap of antimicrobial resistance in *E. coli* isolates collected from Canadian turkey flocks by region in 2018 (n = 367) and 2019 (n = 393).** X-axes represent the antimicrobial classes: amoxicillin (AMC), ampicillin (AMP), azithromycin (AZM), chloramphenicol (CHL), ciprofloxacin (CIP), ceftriaxone (CRO), cefoxitin (FOX), gentamicin (GEN), meropenem (MEM), nalidixic acid (NAL), Sulfisoxazole (SSS), Streptomycin (STR), Trimethoprim-Sulfamethoxazole (SXT), Tetracyclines (TET). Y-axes represent the E. coli isolates included in this study. The blue color depicts susceptibility, and the red color illustrates resistant patterns.
(TIF)

**S3 Fig. A heatmap of antimicrobial resistance in *E. coli* isolates collected from Canadian turkey flocks by region in 2020 (n = 223) and 2021 (n = 429).** X-axes represent the antimicrobial classes: amoxicillin (AMC), ampicillin (AMP), azithromycin (AZM), chloramphenicol (CHL), ciprofloxacin (CIP), ceftriaxone (CRO), cefoxitin (FOX), gentamicin (GEN), meropenem (MEM), nalidixic acid (NAL), Sulfisoxazole (SSS), Streptomycin (STR), Trimethoprim-Sulfamethoxazole (SXT), Tetracyclines (TET). Y-axes represent the E. coli isolates included in this study. The blue color depicts susceptibility, and the red color illustrates resistant patterns.
(TIF)

**S4 Fig. Associations between antimicrobial resistance to individual antimicrobials in *E. coli* isolates (n = 1986) and sampling years and regions determined by the multivariable mixed effect logistic regression analysis.**
(TIF)

**S1 Table. Antimicrobial use numerator and denominator input parameters and number of turkey flocks enrolled in this study from 2016 to 2021.**
(DOCX)

**S2 Table. Summary of the quantity of antimicrobials used (mg/kg) per year by region.**
(DOCX)

**S3 Table. Summary of the AMR in *E. coli* isolates of Canadian turkey flocks per year by region.**
(DOCX)

## Acknowledgments

All authors would like to acknowledge the turkey producers for their consent and participation in the CIPARS Farm Surveillance. We also thank the veterinarians for enabling samples and data collection.

## Author Contributions

**Conceptualization:** Rima D. Shrestha, Csaba Varga.

**Data curation:** Rima D. Shrestha.

**Formal analysis:** Rima D. Shrestha, Csaba Varga.

**Funding acquisition:** Agnes Agunos, Sheryl P. Gow, Csaba Varga.

**Investigation:** Rima D. Shrestha, Agnes Agunos, Anne E. Deckert, Csaba Varga.

**Methodology:** Rima D. Shrestha, Csaba Varga.

**Project administration:** Agnes Agunos, Sheryl P. Gow, Csaba Varga.

**Resources:** Agnes Agunos, Sheryl P. Gow, Anne E. Deckert, Csaba Varga.

**Supervision:** Csaba Varga.

**Validation:** Rima D. Shrestha, Csaba Varga.

**Visualization:** Rima D. Shrestha, Csaba Varga.

**Writing – original draft:** Rima D. Shrestha.

**Writing – review & editing:** Rima D. Shrestha, Agnes Agunos, Sheryl P. Gow, Anne E. Deckert, Csaba Varga.

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
