## [Decision Letter · Decision Letter 0]

28 Nov 2022

PONE-D-22-29703Decrease in the prevalence of antimicrobial resistance in Escherichia coli isolates of Canadian turkey flocks driven by the implementation of an antimicrobial stewardship programPLOS ONE

Dear Dr. Varga,

Thank you for submitting your manuscript to PLOS ONE. After careful consideration, we feel that it has merit but does not fully meet PLOS ONE’s publication criteria as it currently stands. Therefore, we invite you to submit a revised version of the manuscript that addresses the points raised during the review process.

Please respond to all comments by the reviewers.  To cater to readers that are not that statistical savvy, in the materials and methods please include a short 1 or 2 sentence explanation on the use of negative binomial mixed effect regression and then mixed effect multivariable logistic regression models for analysis of the data.  Please keep the name of the statistical technique consistent throughout the manuscript. In the Data Management, include the basis of considering 'intermediate' as susceptible rather than 'resistant. In the Discussion, include discussion of the study findings in relation to the variable AMR findings from countries that have had long term experience of controlling antibiotic usage in food animals such as Scandinavian countries.

We look forward to receiving your revised manuscript.

Kind regards,

Latiffah Hassan

Academic Editor

PLOS ONE

Journal Requirements:

Reviewers' comments:

Reviewer's Responses to Questions

**Comments to the Author**

1. Is the manuscript technically sound, and do the data support the conclusions?

Reviewer #1: Yes

Reviewer #2: Partly

2. Has the statistical analysis been performed appropriately and rigorously? 

Reviewer #1: Yes

Reviewer #2: Yes

3. Have the authors made all data underlying the findings in their manuscript fully available?

Reviewer #1: No

Reviewer #2: No

4. Is the manuscript presented in an intelligible fashion and written in standard English?

Reviewer #1: Yes

Reviewer #2: Yes

5. Review Comments to the Author

Reviewer #1: Rima D. Shrestha（PONE-D-22-29703）and colleagues, examined the AMU and AMR surveillance data over six years (2016-2021), collected by CIPARS from turkey farms across three major turkey-producing Canadian provinces/regions to evaluate the impact of a stepwise AMU reduction strategy on the prevalence of AMR in E. coli isolated from turkey flocks. The topic is highly important in the field for mitigating AMR crisis. The overall study is well organized and analyzed, and I have a few minor comments in the below:

1) considering the data transparency and reproducible result, the raw data or so-called source data should be provided. This is the core part in this study. Therefore, excel sheet with detail information regarding all E. coli AMR data, as well as their metadata, should be provided as the supplemental document. This is the source data needed for further evaluation.

2) Even though the table 2 is informative, I would suggest a few figures based on the data from the table could deliver more attractive trend from the study, I would provide this figures as the supplemental documents.

3) Line 304. It would be nice to group the strains with their regions/provinces, for flocks in this heatmap.

4) Additionally, more informative analysis sould be focus on certain antimicrobials that some flocks use and other flocks do not use (a comparison between two groups). A pari-wise comparison would be more informative, side-by-side comparison, the outcome (decreasing trend) of the use and not use certain antimicrobials. I would like to see these results, which might further support AMU has a direct impact on AMR.

Reviewer #2: The manuscript evaluates the implementation of an antimicrobial stewardship strategy in the Canadian turkey industry. The strategy consist on several stepwise withdrawals of antimicrobial uses; in 2014 3rd generation cephalosporins and fluoroquinolones were withdrawn; in 2018, aminoglycosides, macrolides, penicillins, and streptogramin, and between 2019 and 2020, elimination of preventive use of bacitracin and tetracyclines was implemented. Documenting the effectiveness of antimicrobial stewardship is not a particular novel area, but certainly an important one. The major criticism of this paper is that the claims that the implementation was highly effective is not always supported by the data and/or analysis. The effectiveness of the interventions seems to be based on the year effects using 2016 as referent, but the major implementations took place from 2018 to 2020, so comparing against 2016 does not seem the most useful approach to evaluate interventions (L322-L332). For example, based on your figure 2, it looks like tetracycline resistance started to decrease in 2017 before the step 3 strategy was implemented. What other factors could explain the reduction in resistance?. As tetracycline, gentamicin also indicated a sharp decrease from 2017 to 2018 although the implementation was done at the end of 2018.

If implementation finishes in 2020, is it reasonable to only use data from 2020-2021 as a basis to evaluate the post-implementation period?

Reporting the AMU use (currently in Table 1) as a figure (e.g. histogram of AMU by year and by antibiotic category would help to put the results of AMR in perspective. I.e., Did the reductions in AMU really precede the changes in AMR?

6. PLOS authors have the option to publish the peer review history of their article (what does this mean?). If published, this will include your full peer review and any attached files.

Reviewer #1: **Yes: **Min Yue

Reviewer #2: No

---

## [Author Response · Author response to Decision Letter 0]

7 Jan 2023

1. PLOS ONE Journal Requirements:

Response: We have formatted the manuscript and files to meet the PLOS ONE style requirements.

2. In your Data Availability statement, you have not specified where the minimal data set underlying the results 

described in your manuscript can be found. PLOS defines a study's minimal data set as the underlying data used to reach the conclusions drawn in the manuscript and any additional data required to replicate the reported study findings in their entirety. All PLOS journals require that the minimal data set be made fully available. For more information about our data policy, please see http://journals.plos.org/plosone/s/data-availability.

Response: The minimal dataset is presented in the Supplementary Materials (2016 to 2021 antimicrobial resistance summary and antimicrobial use data; Table S1 & S3).

The AMR data used in the study are available via the Government of Canada, Public Health Agency of Canada’s CIPARS visualization platform that could be accessed via this link: https://www.canada.ca/en/public-health/services/surveillance/canadian-integrated-program-antimicrobial-resistance-surveillance-cipars/interactive-data.html

The AMU data expressed in mg/kg are shown in the supplementary files, organized by year (Table S2).

 

Editor’s specific comments:

To cater to readers that are not that statistical savvy, in the materials and methods please include a short 1 or 2 sentence explanation on the use of negative binomial mixed effect regression and then mixed effect multivariable logistic regression models for analysis of the data. 

Response: We included the following:

“To account for clustering at the turkey-flock-level mixed-effect regression models were constructed by including turkey flocks as random intercepts. Mixed-effects multivariable Poisson regression models were built for the count outcome variable signified by the total antimicrobials used in turkey flocks (AMUtotal-any route) to determine the changes in AMU across years and provinces. Overdispersion of the Poisson multivariable mixed effect model was assessed by applying goodness of fit chi-squared test, and if it was significant, a negative mixed effect binomial model was used.” (Lines: 184-190).

Please keep the name of the statistical technique consistent throughout the manuscript. 

Response: We have reviewed the manuscript and now we use the names of the statistical techniques consistent. 

In the Data Management, include the basis of considering 'intermediate' as susceptible rather than 'resistant. 

Response: We included the following:

“For this study, the intermediate isolates were classified as susceptible following both the NARMS (i.e., any isolate below the clinical resistance breakpoint using CLSI or NARMS/CIPARS interpretative criteria) and the European Committee on Antimicrobial Susceptibility Testing (EUCAST) Steering Committee guidelines [30,31].” (Lines 145-148)

In the Discussion, include discussion of the study findings in relation to the variable AMR findings from countries that have had long term experience of controlling antibiotic usage in food animals such as Scandinavian countries.

Response: We included the following:

“In our study, E. coli isolates had very high resistance to tetracyclines, which is consistent with the previous AMR studies involving the turkey flocks in Canada [23,37], European countries [27,38], and Egypt [39].” (Lines 399-401)

Reviewers' comments:

Reviewer #1 comments:

Rima D. Shrestha（PONE-D-22-29703）and colleagues, examined the AMU and AMR surveillance data over six years (2016-2021), collected by CIPARS from turkey farms across three major turkey-producing Canadian provinces/regions to evaluate the impact of a stepwise AMU reduction strategy on the prevalence of AMR in E. coli isolated from turkey flocks. The topic is highly important in the field for mitigating AMR crisis. The overall study is well organized and analyzed, and I have a few minor comments in the below:

1) considering the data transparency and reproducible result, the raw data or so-called source data should be provided. This is the core part in this study. Therefore, excel sheet with detail information regarding all E. coli AMR data, as well as their metadata, should be provided as the supplemental document. This is the source data needed for further evaluation.

Response: The raw unadjusted AMR data component is available via the CIPARS interactive data platform up to 2019:

https://www.canada.ca/en/public-health/services/surveillance/canadian-integrated-program-antimicrobial-resistance-surveillance-cipars/interactive-data.html

In addition, the minimal dataset is presented in the Supplementary Materials (2016 to 2021 antimicrobial resistance summary and antimicrobial use data; Table S1 & S3).

2) Even though the table 2 is informative, I would suggest a few figures based on the data from the table could deliver more attractive trend from the study, I would provide this figure as the supplemental documents.

Response: Thank you for your suggestion. To illustrate the trend of AMR across years and regions we made a supplementary figure [S1. Fig: Associations between antimicrobial resistance to individual antimicrobials in E. coli isolates (n=1986) and sampling years and regions determined by the multivariable mixed effect logistic regression analysis]

3) Line 304. It would be nice to group the strains with their regions/provinces, for flocks in this heatmap.

Response: We provided a heatmap that shows resistance patterns of E. coli isolates across years and regions. [S2 Fig: A heatmap of antimicrobial resistance in E. coli isolates collected from Canadian turkey flocks by region in 2016 (n=277) and 2017 (n=287)].

4) Additionally, more informative analysis should be focus on certain antimicrobials that some flocks use, and other flocks do not use (a comparison between two groups). A pari-wise comparison would be more informative, side-by-side comparison, the outcome (decreasing trend) of the use and not use certain antimicrobials. I would like to see these results, which might further support AMU has a direct impact on AMR.

Response: This study’s main objective is to show the before and after effects of stepwise reduction/elimination of antimicrobial use (for prevention) on the prevalence of antimicrobial resistance in E. coli isolates. Our previous study has already demonstrated associations between the use of antimicrobials and the emergence of resistance in Canadian turkey flocks (Shrestha RD, Agunos A, Gow SP, Deckert AE, Varga C. Associations between antimicrobial resistance in fecal Escherichia coli isolates and antimicrobial use in Canadian turkey flocks. Front Microbiol. 2022 Jul 29;13:954123. https://doi.org/10.3389/fmicb.2022.954123 )

Reviewer #2 comments:

Reviewer 2: The manuscript evaluates the implementation of an antimicrobial stewardship strategy in the Canadian turkey industry. The strategy consists on several stepwise withdrawals of antimicrobial uses; in 2014 3rd generation cephalosporins and fluoroquinolones were withdrawn; in 2018, aminoglycosides, macrolides, penicillins, and streptogramin, and between 2019 and 2020, elimination of preventive use of bacitracin and tetracyclines was implemented. Documenting the effectiveness of antimicrobial stewardship is not a particular novel area, but certainly an important one. The major criticism of this paper is that the claims that the implementation was highly effective is not always supported by the data and/or analysis. The effectiveness of the interventions seems to be based on the year effects using 2016 as referent, but the major implementations took place from 2018 to 2020, so comparing against 2016 does not seem the most useful approach to evaluate interventions (L322-L332).

Response: We appreciate your comments. The antimicrobial stewardships included stepwise reduction of AMU in the Canadian turkey industry. The implementation of the first step was started in 2014. However, AMR-AMU surveillance was implemented in all five regions of Canada only in 2016. To make it comparable across all regions, we included only the period between 2016 and 2021 and did not use 2013-2015 data. 

To clarify this issue, we included the following: “The CIPARS sentinel farm surveillance started in 2013 in one province of Canada, then gradually expanded to 5 turkey-producing provinces/regions in 2016 to harmonize with other commodities sampled at the farm level. Because of limited geographic coverage in the early phases of the program, this present study included farm data collected between 2016 and 2021.” (Lines 105-108)

Regarding the comparison group, we believe that the effect of the reduction of certain antimicrobials on the prevalence of resistance happens over years and not one year to another. In addition, the reduction of antimicrobials is only for preventative use and farmers still can use antimicrobials to treat bacterial infectious diseases. To clarify this aspect, we included:

“It is important to note that the strategy does not include the prohibition of the use of these antimicrobials for treatment of disease under veterinary supervision in the face of an outbreak.” (Lines 94-96).

Reviewer 2: For example, based on your figure 2, it looks like tetracycline resistance started to decrease in 2017 before the step 3 strategy was implemented. What other factors could explain the reduction in resistance? As tetracycline, gentamicin also indicated a sharp decrease from 2017 to 2018 although the implementation was done at the end of 2018.

Response: To clarify this aspect we included the following:

“This current study examined AMU and AMR in one commensal organism and thus, results should be interpreted with caution. The stepwise antimicrobial reduction/elimination strategy was important in the preservation of the efficacy of the relevant antimicrobials and more importantly, for the containment of AMR, reducing its transmission downstream of the farm-to-fork continuum. As previously described, the antimicrobials targeted by the reduction strategy could still be used for treatment purposes (i.e., during outbreak situations under veterinary supervision), therefore, AMU reduction may not always correspond to an observed reduction in resistance. Simultaneous assessment of other surveillance components (including the full spectrum of the bacterial organism under surveillance) using a One Health approach and evaluation of the animal welfare and economic benefits of the AMU reduction required to understand the implications of the industry-led strategy.” (Lines 476-485)

Reviewer 2: If implementation finishes in 2020, is it reasonable to only use data from 2020-2021 as a basis to evaluate the post-implementation period?

Response: We appreciate your comments. We believe that to show trends over time we need to include several years and not analyze a short period.

Reviewer 2: Reporting the AMU use (currently in Table 1) as a figure (e.g. histogram of AMU by year and by antibiotic category would help to put the results of AMR in perspective. I.e., Did the reductions in AMU really precede the changes in AMR?

Response: Thank you for your valuable suggestion. We included an additional figure (Figure 1. Quantity-based antimicrobial use (AMU) indicators (flock-level AMU in mg/kg turkey biomass) by year.), which shows the use of each antimicrobial quantity over years. In addition, we included a supplementary figure (S1. Fig: Associations between antimicrobial resistance to individual antimicrobials in E. coli isolates (n=1986) and sampling years and regions determined by the multivariable mixed effect logistic regression analysis) that shows the changes in the odds of resistance of E. coli isolates across regions and years.

---

## [Decision Letter · Decision Letter 1]

27 Feb 2023

Decrease in the prevalence of antimicrobial resistance in Escherichia coli isolates of Canadian turkey flocks driven by the implementation of an antimicrobial stewardship program

PONE-D-22-29703R1

Dear Dr. Varga,

We’re pleased to inform you that your manuscript has been judged scientifically suitable for publication and will be formally accepted for publication once it meets all outstanding technical requirements.

Kind regards,

Latiffah Hassan

Academic Editor

PLOS ONE

Additional Editor Comments (optional):

Reviewers' comments:

Reviewer's Responses to Questions

**Comments to the Author**

1. If the authors have adequately addressed your comments raised in a previous round of review and you feel that this manuscript is now acceptable for publication, you may indicate that here to bypass the “Comments to the Author” section, enter your conflict of interest statement in the “Confidential to Editor” section, and submit your "Accept" recommendation.

Reviewer #1: All comments have been addressed

Reviewer #3: All comments have been addressed

2. Is the manuscript technically sound, and do the data support the conclusions?

Reviewer #1: Yes

Reviewer #3: Yes

3. Has the statistical analysis been performed appropriately and rigorously? 

Reviewer #1: Yes

Reviewer #3: Yes

4. Have the authors made all data underlying the findings in their manuscript fully available?

Reviewer #1: Yes

Reviewer #3: Yes

5. Is the manuscript presented in an intelligible fashion and written in standard English?

Reviewer #1: Yes

Reviewer #3: Yes

6. Review Comments to the Author

Reviewer #1: none, I feel that great improvements from current version, which is ready for conditionally acceptance.

Reviewer #3: The authors have satisfactorily addressed comments from the reviewers. However, the following issues should be addressed to improve the clarity of the manuscript:

-Line 204: Include the percentages for the proportions of the farms that used antimicrobials by region (Western region 204 (n=157/209,?), followed by Ontario (n=100/153, ?) and Québec (n=75/148,?) so that the reporting formats are consistent with those presented before ( see Lines 201-203).

-Use commas (,) to separate thousands in figures with four (4) digits and above so that these figures are not confused with years in reference. For example, use 1,976 in place of 1976. Do this for all 4-digits and above figures throughout the manuscript.

-Line 396: insert a space between ‘Quebec’ and ‘and’ to show that they are two separate words.

7. PLOS authors have the option to publish the peer review history of their article (what does this mean?). If published, this will include your full peer review and any attached files.

Reviewer #1: **Yes: **Min Yue

Reviewer #3: **Yes: **Hassan Ismail Musa

---

## [Editor Report · Acceptance letter]

4 Apr 2023

PONE-D-22-29703R1 

Decrease in the prevalence of antimicrobial resistance in *Escherichia coli* isolates of Canadian turkey flocks driven by the implementation of an antimicrobial stewardship program 

Dear Dr. Varga:

I'm pleased to inform you that your manuscript has been deemed suitable for publication in PLOS ONE. Congratulations! Your manuscript is now with our production department. 

Kind regards, 

on behalf of

Prof Dr. Latiffah Hassan 

Academic Editor

PLOS ONE